



# A dataset of tracer concentrations and meteorological observations from the Bolzano Tracer EXperiment (BTEX) to characterize pollutant dispersion processes in an Alpine valley.

Marco Falocchi[1,2], Werner Tirler[3], Lorenzo Giovannini[1], Elena Tomasi[4], Gianluca Antonacci[4], and Dino Zardi[1,2]

[1]Atmospheric Physics Group, Department of Civil Environmental and Mechanical Engineering, University of Trento, Trento (Italy)
[2]C3A – Center Agriculture Food Environment, University of Trento, Trento (Italy)
[3]Eco–Research srl, Via Negrelli 13, 39100 Bolzano (Italy)
[4]CISMA S.r.l., Via Ipazia 2 c/o NOI Techpark, 39100 Bolzano (Italy)

**Correspondence:** Marco Falocchi (marco.falocchi@unitn.it)

**Abstract.** The paper describes the dataset of concentrations and related meteorological measurements collected during the field campaign of the Bolzano Tracer Experiment (BTEX). The experiment was performed to characterize the dispersion of pollutants emitted from a waste incinerator in the basin of the city of Bolzano, in the Italian Alps. As part of the experiment two controlled releases of a passive gas tracer (sulfure exafluoride, $SF_6$) were performed through the stack of the incinerator

5 on 14 February 2017 for two different time–lags, starting respectively at 07:00 LST and 12:45 LST. Samples of ambient air were collected at target sites with vacuum–filled glass bottles and polyvinyl fluoride bags, and later analyzed by means of a mass spectrometer (detectability limit $30\,\mathrm{ppt_v}$). Meteorological conditions were monitored by a network of 15 ground weather stations, 1 microwave temperature profiler, 1 SODAR and 1 Doppler Wind–LIDAR.

The dataset represents one of the few examples available in the literature concerning dispersion processes in a typical

10 mountain valley environment and provides a useful benchmark for testing atmospheric dispersion models in complex terrain. The dataset described in this paper is available at https://doi.pangaea.de/10.1594/PANGAEA.898761 (Falocchi *et al.*, 2019).

## 1 Introduction

Pollutant transport modeling is an essential tool for our understanding of factors controlling air quality and consequently affecting the environment and the human health. Nowadays, the increasing computational capabilities allow to simulate with

15 unprecedent detail many atmospheric processes even at local scale. However, models still need careful calibration and validation against field observations, especially over mountainous complex terrain, where the interaction between local atmospheric processes and the orography (*c.f.* Zardi and Whiteman, 2013; Serafin et al., 2018) affects both mean flow and turbulence properties (*c.f.* Rotach and Zardi, 2007; Giovannini et al., 2019), and therefore makes the advection and dispersion of pollutants further complicated (*e.g.* De Wekker et al., 2018; Tomasi et al., 2018). Moreover, the frequent occurrence of persistent low–





level inversions and cold pools in depressed areas and closed basins may inhibit the dispersion of pollutants, especially during wintertime (*e.g.* de Franceschi and Zardi, 2009).

Several research projects focusing on air pollution transport processes at different scales performed experiments including controlled releases of tracers, both at ground level (*e.g.* Cramer et al., 1958; Doran and Horst, 1985) and from elevated sources (*e.g.* Nieuwstadt and van Duuren, 1979), such as the stack of industrial plants (*e.g.* Emberlin, 1981; Zannetti et al., 1986; Hanna et al., 1986). For example, Min et al. (2002) investigated the dispersion processes above the atmospheric boundary layer by releasing puffs of sulfur hexafluoride ($SF_6$), quantifying the dispersion by analysing images provided by multiple infrared cameras placed at the ground. The European tracer experiment (ETEX, Van dop et al., 1998; Nodop et al., 1998), instead, focused on the assessment of the ability of various dispersion models in simulating emergency response situations across northern Europe. The critical events were emulated by performing two controlled releases of a passive gas tracer about 35 km west of Rennes (France). Model predictions were then validated against measurements of tracer concentration at 168 surface stations in 17 different countries. Field campaigns involving releases of passive tracers in cities, from both elevated (*e.g.* Camuffo et al., 1979; Gryning and Lyck, 1980; Allwine and Flaherty, 2006; Doran et al., 2007) and surface sources (*e.g.* Britter et al., 2002; Gryning and Lyck, 2002; Clawson et al., 2005; Gromke et al., 2008; Wood et al., 2009), were also carried out to identify how pollutants migrate within the urban environment, and to identify the areas of the cities where the population is mostly exposed to high concentrations.

The main purpose of the above experiments is to provide a reliable reference benchmark under controlled conditions for comparison with dispersion model outputs. However, only few projects were conducted over complex mountainous terrain (*e.g.* Whiteman, 1989; Ambrosetti et al., 1994; Kalthoff et al., 2000; Darby et al., 2006), due to the intrinsic difficulties both in designing and in managing measurement campaigns in this specific environment, and in modeling local meteorological processes (*e.g.* Giovannini et al., 2014, 2017; Serafin et al., 2018; De Wekker et al., 2018).

The present paper describes a dataset of tracer concentrations and related meteorological measurements collected to evaluate the pollutant dispersion from a waste incinerator close to Bolzano, a mid–sized city in the Italian Alps. Indeed, concerns about the environmental impacts of this plant stimulated the organization of a comprehensive project to assess the fate of the pollutants released and their ground deposition in the surrounding area (Ragazzi et al., 2013). The project, called BTEX (Bolzano Tracer EXperiment), was then designed and performed. The project included a measurement campaign with tracer releases from the incinerator stack, with the final aim of validating the emission–impact scenarios simulated by a modeling chain composed of meteorological and dispersion models (Tomasi et al., 2019). The controlled releases of the tracer were performed on 14 February 2017, with the plant working at operating speed, at two different times of the day, in order to investigate the dispersion processes under different conditions of atmospheric stability and blowing winds. During the measurement campaign, the meteorological variables in the basin were monitored by means of 15 surface weather stations, one microwave temperature profiler, one SODAR and one Doppler Wind–LIDAR.

The paper is organized as follows: Sec. 2 describes the outline of the BTEX project, Sec. 3 provides details of the meteorological monitoring network and of the tracer concentration measurement techniques. The collected dataset, instead, is described in Sec. 4 and discussed in Sec. 5. Finally, conclusions and outlooks are drawn in Sec. 6.

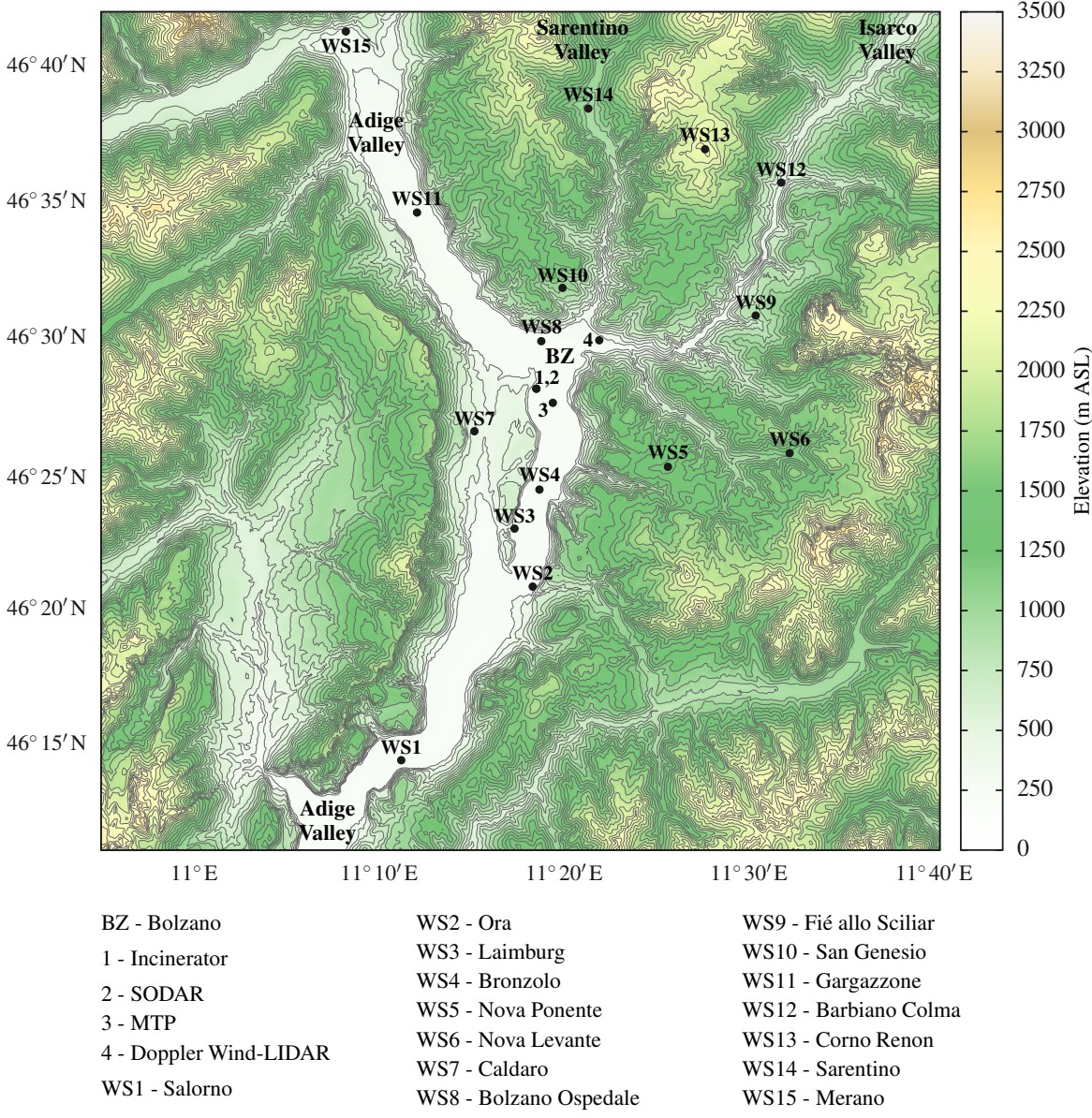

BZ - Bolzano

1 - Incinerator

2 - SODAR

3 - MTP

4 - Doppler Wind-LIDAR

WS1 - Salorno

WS2 - Ora
WS3 - Laimburg
WS4 - Bronzolo
WS5 - Nova Ponente
WS6 - Nova Levante
WS7 - Caldaro
WS8 - Bolzano Ospedale

WS9 - Fié allo Sciliar
WS10 - San Genesio
WS11 - Gargazzone
WS12 - Barbiano Colma
WS13 - Corno Renon
WS14 - Sarentino
WS15 - Merano

**Figure 1.** Overview of the study area. The map shows the orographic complexity of the region, the position of the city of Bolzano (BZ), of the incinerator (1) and of the meteorological monitoring used during BTEX, namely one SODAR (2), one microwave temperature profiler (MTP, 3), one Doppler Wind–LIDAR (4) and 15 ground weather stations (WS).





## 2   Outline of the BTEX project

### 2.1   Target area

The city of Bolzano ($258\,\mathrm{m}\,\mathrm{ASL}$, northeastern Italian Alps), where about 107000 inhabitants live, is the most populated town of South Tyrol. As shown in Fig. 1, the city lies in a wide basin at the junction of the Adige Valley (S and NW) with the Sarentina Valley (N) and the Isarco Valley (E). The area is characterized by a highly complex mountainous topography, with very steep sidewalls and many surrounding crests exceeding $1200\,\mathrm{m}\,\mathrm{ASL}$. Such a morphology deeply affects the local meteorology. Indeed, the elevated ridges allow at lower levels the onset of wind regimes mainly dominated by thermally–driven local circulations, sometimes associated to the development of ground–based temperature inversions. Moreover, especially under fair weather conditions, the nocturnal wind regime inside the basin is mostly dominated by drainage winds from the Isarco Valley, accelerating at the outlet of the valley and spreading into the Bolzano basin like a valley–exit jet, with peaks of intensity exceeding $12\,\mathrm{m}\,\mathrm{s}^{-1}$ (Tomasi et al., 2019). As it is well documented in the literature (*e.g.* Whiteman, 2000; Zardi and Whiteman, 2013), thermally–driven valley–exit jets develop when a narrow valley abruptly joins the adjacent plain. Indeed, the Isarco Valley displays these characteristics (see in Fig. 1 the stretch from Barbiano Colma (WS12) to the outlet).

The waste incinerator is $2\,\mathrm{km}$ southwest of Bolzano, in the Adige Valley (label 1 in Fig. 1) and became operative since July 2013. The plant has a maximum treatment capacity of $130000\,\mathrm{t}\,\mathrm{y}^{-1}$ and was designed to treat the exhaust–smokes along a three stage system, before they get released in the atmosphere at $60\,\mathrm{m}\,\mathrm{AGL}$, with a temperature of $413\,\mathrm{K}$ and at a rate of $10^5\,\mathrm{Nm}^3\,\mathrm{h}^{-1}$.

### 2.2   Experimental set up

#### 2.2.1   Tracer selection

A passive tracer for monitoring the dispersion processes in the atmosphere needs to be: (*i*) colourless and odourless; (*ii*) non–toxic for the human health and the environment; (*iii*) chemically inert and stable at the temperature of the released smoke; (*iv*) absent in the ambient air (*i.e.* detectable only in trace concentration); and (*v*) easily measurable in the laboratory once environmental samples have been collected. As none of the substances normally released in the atmosphere by the plant fulfill the above requirements, sulfur hexafluoride ($SF_6$, $MM = 146\,\mathrm{g}\,\mathrm{mol}^{-1}$) was selected as the passive tracer for BTEX. Indeed, this gas is not naturally present in the ambient air (background around $10\,\mathrm{ppt_v}$ at global level, *e.g.* Rigby et al., 2010; Manca, 2017) and has been widely used in many other studies concerning the dispersion of pollutants from the stack of industrial plants (*e.g.* Emberlin, 1981; Bowne et al., 1983; Zannetti et al., 1986; Sivertsen, 1988), as well as atmospheric dispersion processes in mountain valleys (*e.g.* Whiteman, 1989; Kalthoff et al., 2000; Darby et al., 2006) and in the urban environment (*e.g.* Camuffo et al., 1979; Gryning and Lyck, 1980; Britter et al., 2002; Allwine and Flaherty, 2006; Doran et al., 2007; Gromke et al., 2008; Wood et al., 2009).

Nowadays, $SF_6$ is mostly adopted in industry as an electrical insulator. Leakages of $SF_6$ from plants represent the most important source likely to alter the background concentration. Accordingly, due to the many industrial activities in the surround-





ings of Bolzano, a preliminary campaign was performed to investigate the background concentration of $SF_6$ in the ambient air of the Bolzano basin prior to any tracer release. Concentrations were found to be lower than $30\,ppt_v$ (instrumental detection limit, see Sec. 3.2 for the laboratory detection method) and therefore comparable with the global background concentration.

### 2.2.2 Tracer injection strategy

Pollutant emissions from a plant into the atmosphere are strongly controlled by the operational working of the plant. The emissions from the incinerator of Bolzano, under usual operating conditions, are ejected at a constant flow rate under steady conditions. Therefore, the deposition areas of the pollutants and their ground concentrations mostly depend on local atmospheric processes, *e.g.* wind regime and stability. Accordingly, in order to obtain realistic deposition patterns of the tracer, an important requirement was to perform a steady release of $SF_6$ at constant concentration. In particular, $SF_6$ was injected at the bottom of the stack, upstream of the ventilation system, to guarantee an effective mixing of the tracer within the smoke, having the incinerator working at operational speed. The control on the mass released, instead, was performed by regulating the valves of the tank containing the tracer. To avoid tracer gas condensation (as a consequence of the expansion) and related consequences on the injection system, possibly resulting in a non–constant gas injection, a monitoring system was installed, based on an on–line mass spectrometer to measure the actual $SF_6$ concentration in the smoke before the outlet into the atmosphere. In particular, an on–line mass spectrometer with EI ionization, manifactured by V&F Instruments (Airsense Compact), was used. This ionization mode allows to achieve a relatively low sensitivity only, which is nonetheless enough to monitor continuously the high $SF_6$ concentrations in the smoke (several hundreds of ppm).

### 2.2.3 Strategy of the releases

Two releases of $SF_6$ were performed on 14 February 2017, in order to investigate the dispersion processes under the absence of significant synoptic forcing, but with different conditions of atmospheric stability and wind direction. Indeed, as detailed in Sec. 5, the first release was planned in the early morning, at 07:00 LST, when the atmosphere was stable and a weak down–valley wind was blowing in the Adige Valley. The second release, instead, was scheduled in the early afternoon, at 12:45 LST, with a weakly unstable atmosphere and an up–valley wind blowing.

### 2.2.4 Sampling strategy

After each release, samples of ambient air were collected by means of vacuum–filled glass bottles and polyvinyl fluoride (PVF) bags (hereafter PVF bags). Bottles with a capacity of $1\,liter$ were filled by means of a valve, whose calibrated nozzle allowed a constant inflow, thus determining the time required to fill the bottle. In the framework of BTEX, nozzles were calibrated to fill a bottle in 20 or 60 minutes. The filling of the PVF bags, instead, was regulated by a suitable air pump controlling the inflow speed.

During each release of $SF_6$, 14 sampling teams were located into the field. A crucial issue in the design of the sampling strategy was represented by the definition of the sampling–point positions and their scheduling, *i.e.* starting time and duration.



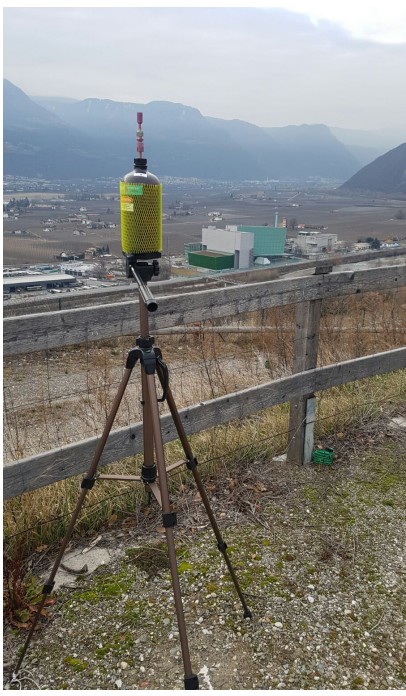 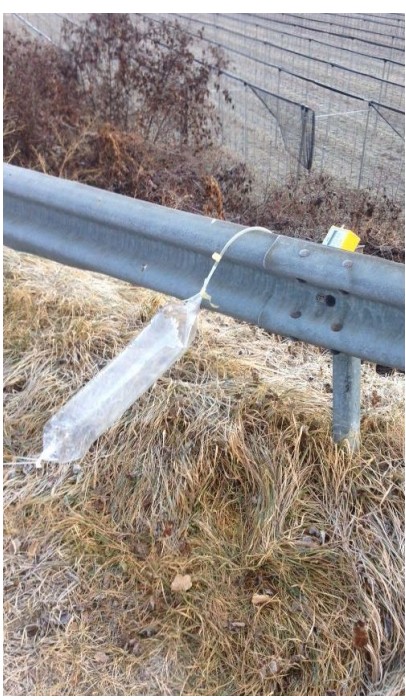

**Figure 2.** Vacuum–filled glass bottle (left) and PVF bag (right) used to collect samples of ambient air. These pictures were taken during the tracer releases of BTEX. On the right of the vacuum bottle, the incinerator of Bolzano can be recognised.

Given the complex orography of the study area and its related heterogeneous meteorological fields, the sampler distribution could be neither geometrically designed (as for example with concentric circles around the incinerator stack, *e.g.* Hanna et al., 1986) nor fixed on the basis of the mean wind flow. For these reasons, seven sampling teams were located in the main residential areas, where most of the receptors are concentrated, while the remaining ones were placed in the surroundings of the incinerator, in agreement with the ground concentrations foreseen by an operational modeling chain composed of both meteorological and air quality models, specifically set–up for the present field campaign. A detailed description of this modeling chain is beyond the purposes of the current paper and will be presented in future related works. Here, its structure and operational use during the experimental campaigns is briefly reported.

The meteorological field inside the Bolzano basin was simulated by means of the Weather and Research Forecasting (WRF) model (Skamarock and Klemp, 2008) through four nested domains, with an increasing horizontal resolution of $13.5\,\mathrm{km}$ (North Italy), $4.5\,\mathrm{km}$ (Northeastern Italian Alps), $1.5\,\mathrm{km}$ (Trentino Alto Adige) and $0.5\,\mathrm{km}$ (Bolzano basin). Moreover, in the innermost domain, observational nudging was adopted, by assimilating all the available data described in Sec. 3.1. Then, the CALMET model was used to increase the horizontal resolution of the meteorological field simulated by WRF up to $200\,\mathrm{m}$, whereas the CALPUFF model was adopted to simulate the dispersion processes and the ground concentrations within the study area.

The modeling chain was used:





**Table 1.** Surface weather stations (WS) adopted to monitor the meteorological field during BTEX, along with their position (V: valley floor station; S: valley sidewall station), coordinates, altitude and available measurements: air temperature (T); relative humidity (RH); rain (R); atmospheric pressure (P); wind direction (dir); wind intensity (U); wind gust ($U_g$); global radiation (GR); sunshine duration (SD). The first column indicates the label of the station shown in Fig. 1.

| | Station | | | Coordinates | | Altitude | Meteorological quantities | | | | | | | | |
| | Site | ID | Position | lat. [N] | lon. [E] | [m ASL] | T | RH | R | P | dir | U | $U_g$ | GR | SD |
|---|---|---|---|---|---|---|---|---|---|---|---|---|---|---|---|
| WS1 | SALORNO | 88820MS | V | 46.24 | 11.19 | 212 | x | x | x | x | x | x | x | x | x |
| WS2 | ORA | 86900MS | V | 46.35 | 11.30 | 250 | x | x | x | x | | | | x | x |
| WS3 | LAIMBURG | 86600MS | V | 46.38 | 11.29 | 224 | x | x | x | x | x | x | x | x | x |
| WS4 | BRONZOLO | 85700MS | V | 46.41 | 11.31 | 226 | x | x | x | x | x | x | x | x | x |
| WS5 | NOVA PONENTE | 85120MS | S | 46.42 | 11.43 | 1470 | x | x | x | x | x | x | x | x | x |
| WS6 | NOVA LEVANTE | 78305MS | S | 46.43 | 11.54 | 1128 | x | x | x | x | | | | x | x |
| WS7 | CALDARO | 89190MS | V | 46.44 | 11.25 | 495 | x | x | x | x | x | x | x | x | x |
| WS8 | BOLZANO OSPEDALE | 83200MS | V | 46.50 | 11.31 | 254 | x | x | x | x | x | x | x | x | x |
| WS9 | FIÉ ALLO SCILIAR | 75600MS | S | 46.51 | 11.51 | 840 | x | x | x | x | x | x | x | x | x |
| WS10 | SAN GENESIO | 82910MS | S | 46.53 | 11.33 | 970 | x | x | x | x | x | x | x | x | x |
| WS11 | GARGAZZONE | 27100MS | V | 46.58 | 11.20 | 290 | x | x | x | x | x | x | x | x | x |
| WS12 | BARBIANO COLMA | 74900MS | V | 46.60 | 11.53 | 490 | x | x | x | x | x | x | x | x | x |
| WS13 | CORNO RENON | 82500WS | S | 46.62 | 11.46 | 2260 | x | x | x | | | | | | |
| WS14 | SARENTINO | 82200MS | S | 46.64 | 11.36 | 970 | x | x | x | x | x | x | x | x | x |
| WS15 | MERANO | 23200MS | V | 46.69 | 11.14 | 330 | x | x | x | x | x | x | x | x | x |

1. 48 h before the experiment, to pinpoint a "first–guess" distribution of the samplers on the basis of the predicted meteorological conditions and fall–out areas.

2. Right before each release, in nowcasting mode, to adjust (if needed) the distribution of the samplers with the updated model output (driven by the most recent assimilated meteorological measurements) and to define the starting time and duration of each sampling.

## 3 Materials and methods

### 3.1 Meteorological measurements

During BTEX, the meteorological field within the study area was monitored by means of 15 surface weather stations (WS in Fig. 1), one Microwave Temperature Profiler (MTP, label 3 in Fig. 1), one SODAR (label 2 in Fig. 1) and one Doppler Wind–LIDAR (label 4 in Fig. 1).





**Table 2.** Technical characteristics of the MTP–5HE installed at the airfield of Bolzano.

| Characteristic | Value |
|---|---|
| Site | Bolzano airport (238 m ASL) |
| Elevation | 12 m AGL |
| Field of view azimuth angle | South |
| Range | 1000 m |
| Effective resolution | 50 m from 0 to 100 m |
| | 70 m from 100 to 400 m |
| | 80 m from 400 to 600 m |
| | 120 m from 600 to 1000 m |
| Displayed resolution | 50 m |
| Accuracy under adiabatic conditions | $0.3^\circ$C from 0 to 500 m |
| | $0.4^\circ$C from 500 to 1000 m |
| Accuracy under inversion conditions | $0.8^\circ$C from 0 to 500 m |
| | $1.2^\circ$C from 500 to 1000 m |
| Azimuth direction of the scan | South |
| Number of scans and zenith range | 11, from $0^\circ$ to $90^\circ$ |
| Opening angle | $6^\circ$ |
| Duration of a measurement | 120 s |
| Acquisition rate | 1 profile every 10 min |
| Calibration and diagnostic | Automatic |

### 3.1.1 Surface weather stations

The 15 surface weather stations are part of the network operated by the Meteorological Service of the Autonomous Province of Bolzano, and are partly located on the valley floors and partly on the sidewalls. These stations, listed in Tab. 1 and reported in Fig. 1, record every 10 min the following meteorological quantities: air temperature and humidity at 2 m AGL, average wind intensity and direction and wind gusts at 10 m AGL, rainfall, atmospheric pressure, global radiation and sunshine duration.

### 3.1.2 Microwave temperature profiler

The atmospheric thermal structure inside the basin is constantly monitored by means of an MTP–5HE passive microwave radiometer (manifactured by Attex, Russia). The instrument determines the vertical temperature profile at 11 elevation angles (from $0^\circ$ to $90^\circ$), by measuring the brightness temperature of the molecular oxygen at frequency of 60 GHz. The device retrieves the air temperature from the measured brightness temperature by solving a Fredholm equation of the first kind (Gaikovich et al., 1992; Troitskii et al., 1993) and interpolates the obtained profile along 21 vertical levels $50-$m spaced. Further information





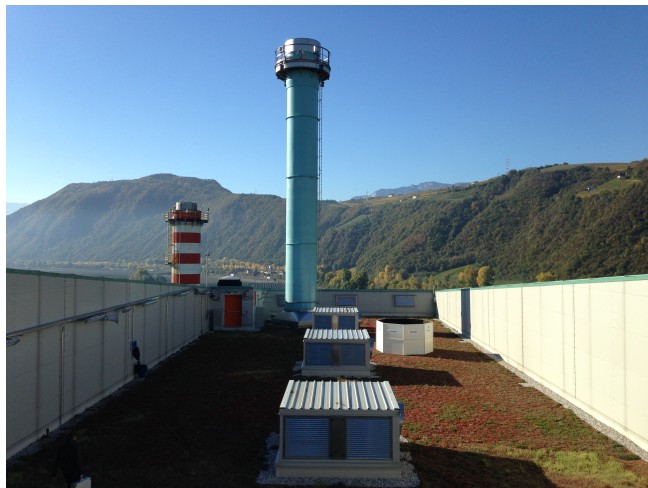
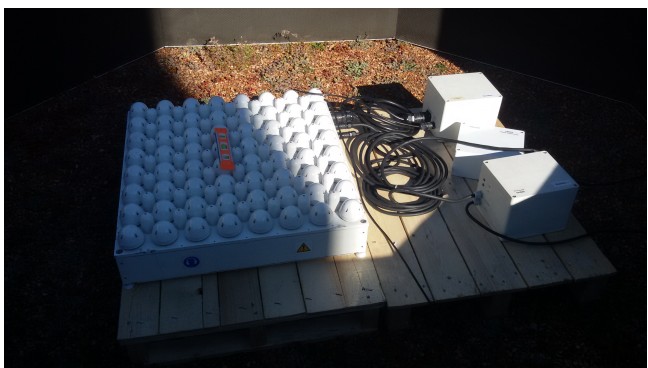

**Figure 3.** A view on the roof of the incinerator of Bolzano during the experimental campaign with the MFAS mini–SODAR (left) and the detail of the antenna (right).

on the functioning of this device can be found in Westwater et al. (1999), Scanzani (2010), Ezau et al. (2013) and Wolf et al. (2014).

The MTP–5HE used during BTEX (technical specifications in Tab. 2, label 3 in Fig. 1), is operated by the Physical Chemistry Laboratory of the Environmental Protection Agency of the Autonomous Province of Bolzano and is installed at the airfield of Bolzano (46.46 N, 11.32 E, 238 m ASL), on the roof of a $12-$m–high building. This device provides every 10 min the vertical

temperature profile from 250 m ASL to 1250 m ASL.

### 3.1.3 SODAR

Wind intensity and direction at the outlet of a stack represent a fundamental input to properly model the fate of pollutants. Accordingly, the wind field at the chimney of the incinerator was monitored by means of a MFAS mini–SODAR (manufactured by Scintec, Germany), installed on the roof of the plant at 40 m AGL (label 2 in Fig. 1). The SODAR, shown in Fig. 3, probed

the wind field within a $400-$m–thick atmospheric layer, from 315 m ASL to 685 m ASL and with a vertical resolution of 30 m. Raw data were processed by means of the software APRun, provided by Scintec, in order to obtain the vertical profiles of the wind speed components and of the backscatter over an averaging period of 30 min. In particular, the profiles were provided along 38 vertical levels, each spaced by 10 m.

SODAR measurements played a key role in the BTEX project. Indeed, during preliminary analyses preceding the experi-

ment, these measurements allowed to capture an intense nocturnal wind (wind intensity greater than $10 \, \text{m s}^{-1}$) blowing from north-east. These observations, supported by high–resolution numerical simulations run with the WRF model, allowed to identify this wind as the drainage flow of the Isarco Valley that, especially under fair weather conditions, spreads into the Bolzano basin behaving like a thermally–driven valley–exit jet (Whiteman, 2000).





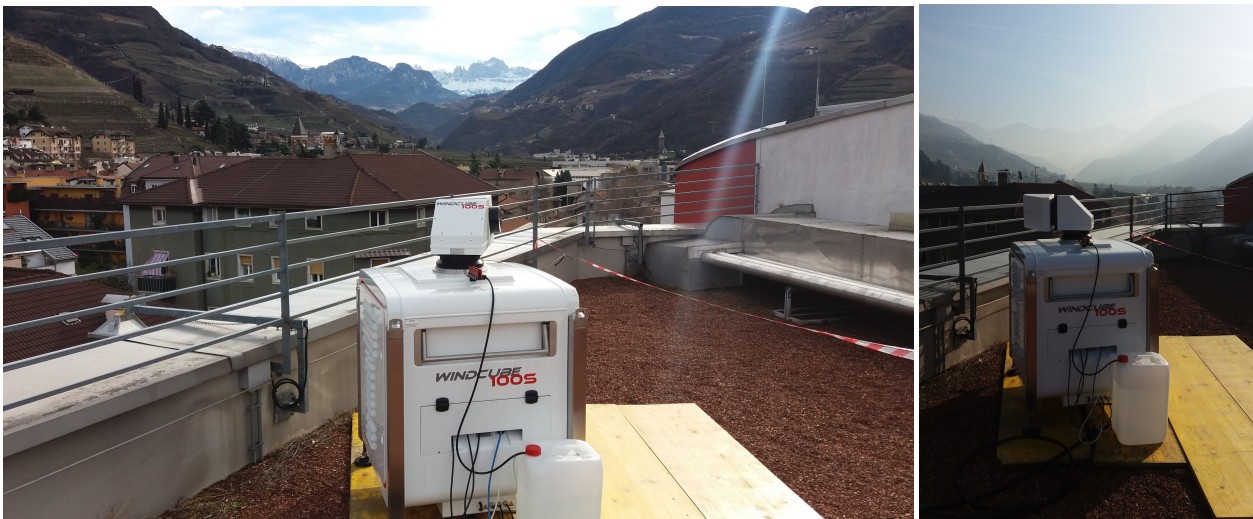

**Figure 4.** The WINDCUBE 100S Doppler Wind–LIDAR from Leosphere (France) installed on a roof at $18\,\mathrm{m\,AGL}$. In the background the outlet of the Isarco Valley.

### 3.1.4 Doppler Wind–LIDAR

The interaction between the valley–exit jet of the Isarco Valley and the smokes of the incinerator can strongly affect the pollutant dispersion scenarios within the Bolzano basin. Therefore, a dedicated measurement campaign was performed to monitor this flow and a Doppler Wind–LIDAR (label 4 in Fig. 1) was installed, from 9 January to 5 April 2017, on the roof of a public building at $18\,\mathrm{m\,AGL}$, at the outlet of the Isarco Valley (Fig. 4).

The Doppler Wind–LIDAR was a WINDCUBE 100S manifactured by Leosphere (France). This device probes the atmo-
sphere with coherent beams of pulsed, eye–safe electromagnetic waves in the near–infrared domain (wavelength $1.54\,\mu\mathrm{m}$) and returns measurements of the wind speed component along each line of sight with physical resolutions of 25, 50, 75 or $100\,\mathrm{m}$, up to 319 gates. In the framework of BTEX, the vertical profile of the wind was measured every $5\,\mathrm{s}$ by means of the Doppler Beam Swinging (DBS) technique along 110 vertical levels, $10\,\mathrm{m}$-spaced (from $335\,\mathrm{m\,ASL}$ to $1385\,\mathrm{m\,ASL}$) and with a vertical resolution of $25\,\mathrm{m}$. These profiles were hourly averaged and assimilated by the modeling system to nowcast both the
meteorological variables and the tracer dispersion scenarios.

### 3.2 Tracer concentration measurements

Differently from the on–line monitoring of $SF_6$ at the incinerator stack, a much higher sensitivity was required for the detection of the tracer concentrations in the samples of ambient air collected in the surroundings of the plant. Accordingly, a triple quadrupole Gas–Chromatography Mass Spectrometer (TSQ 8000 EVO from Thermo Scientific) with chemical ionization with
methane as reagent gas was used in the laboratory analyses. The obtained sensitivity allowed to measure down to background





levels. Nevertheless, to avoid the influence of the background (see Sec. 2.2.1), $SF_6$ concentrations in the collected samples were quantified at levels above $30\,ppt_v$.

## 4   Dataset

The dataset presented in this paper consists of:

 – 79 samples of $SF_6$ concentrations measured at ground level. According to the descriptions provided in Sections 2.2.2, 2.2.4, 2.2.3 and 3.2, the volumes of ambient air were collected in 14 different points surrounding the incinerator for the two releases performed during BTEX.

 – Time series of meteorological quantities collected by the monitoring network described in Sec. 3.1. The 15 surface weather stations contributing to the meteorological data set are listed in Table 1, along with their identification label,
     coordinates, altitude and available observations (x). These data are provided with a time resolution of $10\,min$ and can also be freely downloaded from the open data web–service of the Autonomous Province of Bolzano.

 – Temperature profiles collected every $10\,min$ and provided by the Physical Chemistry Laboratory of the Environmental Agency of the Autonomous Province of Bolzano.

 – SODAR measurements, *i.e.* vertical profiles of mean wind quantities, standard deviations and backscatter, provided every
     $30\,min$.

 – Doppler Wind–LIDAR vertical profiles of the wind speed components and of the Carrier–to–Noise Ratio, provided every $5\,s$.

 – Information and working conditions of the incinerator observed during the day of the release, *i.e.* coordinates and height of the stack, discharge and temperature of the smokes every $30\,min$. These data can also be freely downloaded from the
     web page of the company operating the plant, Eco–Center S.p.A.

## 5   Results and discussions

### 5.1   Weather conditions

Releases of $SF_6$ were performed on 14 February 2017, when over North Italy a high–pressure system was present with very weak synoptic winds (see maps of geopotential height at $500\,hPa$ in Fig. 5). On 13 February, instead, the weakening of a
low–pressure system above the Iberian Peninsula induced southwestern synoptic winds blowing over the study area, which channelled into the valleys. Especially during the afternoon and the evening of 13 February, these synoptic winds interacted with the local circulations by strengthening the up–valley winds, as observed by the surface weather stations (Fig. 6a) and by both the SODAR and the Doppler Wind–LIDAR (Fig. 7b–e). As a further effect, the synoptic winds transported moist air into





**Figure 5.** CFS reanalyzes of geopotential height at $500\,hPa$ from 20170213 00 UTC to 20170215 00 UTC (www.wetterzentrale.de).

the valleys. Indeed, the distributions of absolute humidity shown in Fig. 6c display an increase of the moisture content in the

afternoon of 13 February at all the weather stations used in BTEX (Tab. 1).

During the night between 13 and 14 February, the moist air produced low–level clouds that covered the sky above the study area and inhibited the radiative cooling of the ground. Indeed, as confirmed by the microwave temperature profiler (Fig. 7a), this night was warmer than the previous one, and no significant ground–based temperature inversions developed. The absence of synoptic forcing, instead, allowed the development of thermally–driven circulations. However, the wind regimes captured

by the surface weather stations on the floor of the Adige Valley at Bronzolo (WS4 in Fig. 1) and at Gargazzone (WS11, Fig. 1) appear different from the one observed in the Isarco Valley at Barbiano Colma (WS12 in Fig. 1). Indeed, as shown in Fig.



**Table 3.** Summary of the two releases performed during BTEX (VB: vacuum–filled glass bottles, TB: PVF bags)

|     | Release |  |  | Plant smokes |  | Samples |  |  |
| --- | --- | --- | --- | --- | --- | --- | --- | --- |
| N. | Start [LST] | Duration | Tracer mass | Temperature | Exit speed | Points | VB | TB |
| 1$^{st}$ | 07:00 | 60 min | 150 kg | 140°C | 7.9 m s$^{-1}$ | 14 | 25 | 3 |
| 2$^{nd}$ | 12:45 | 90 min | 450 kg | 140°C | 7.8 m s$^{-1}$ | 14 | 30 | 21 |

6a and b, the winds observed at Bronzolo (WS4) and Gargazzone (WS11) were weak and with intensities less than $1 \, \mathrm{m \, s^{-1}}$, while at Barbiano Colma (WS12) the down–valley wind was much stronger, with intensities around $3 \, \mathrm{m \, s^{-1}}$. In particular, the topographic constraints of the Isarco Valley and of its outlet into the Bolzano basin determined the acceleration of the winds

blowing from this valley, thus allowing the development of a thermally–driven valley–exit jet, whose evolution was observed by the Doppler Wind–LIDAR. The time–height plots of the wind intensity (Fig. 7d) and of the wind direction (Fig. 7e) measured by the Doppler Wind–LIDAR reveal the valley–exit jet blowing from East after 21:00 LST (UTC+1) with intensities around $5 \, \mathrm{m \, s^{-1}}$ in a layer about 400-m deep. During the night, the wind became stronger on an increasingly deep layer until the early morning (14 Feburary 09:00 LST), when it reached its maximum intensity, exceeding $13 \, \mathrm{m \, s^{-1}}$. Then, as it is typical of these

phenomena (Banta et al., 1995; Chrust et al., 2013), the jet abruptly ceased around 11:00 LST. The valley–exit jet was also observed at ground level by the weather stations at Bolzano Ospedale (WS8 in Fig. 1) and Caldaro (WS7 in Fig. 1). Indeed, in the morning of 14 February, the weather station at Bolzano Ospedale (WS8, Fig. 6a, b) recorded a strong easterly wind exceeding $5 \, \mathrm{m \, s^{-1}}$. The weather station at Caldaro (WS7, Fig. 6a, b), instead, measured a northerly wind around $3 \, \mathrm{m \, s^{-1}}$ that intensified from 06:00 LST to 09:00 LST (wind speed above $4 \, \mathrm{m \, s^{-1}}$), in agreement with the behaviour of the valley–exit jet.

The footprint of the valley–exit jet can also be observed in the time series of absolute humidity (Fig. 6c) measured by the previously mentioned surface weather stations, *i.e.* Barbiano Colma (WS12), Bolzano Ospedale (WS8) and Caldaro (WS7). Indeed, in these series, a significant reduction of absolute humidity is observed during the valley–exit jet event. Therefore, it seems reasonable to assume that the valley–exit jet advected drier air from the upper valleys into the basin. After the noon of 14 February, both the SODAR and the Doppler Wind–LIDAR observed a weak up–valley wind (around $2 \, \mathrm{m \, s^{-1}}$) rising from

South in the Adige Valley. Also at ground level, the surface weather stations (Fig. 6a and b) recorded very weak winds, with intensities around $1 \, \mathrm{m \, s^{-1}}$.





**Figure 6.** Time evolution of the wind intensity (a), of the wind direction (b) and the absolute humidity (c) measured by the 15 ground weather stations used during BTEX (see Tab. 1). The absolute humidity was computed from data of relative humidity, air temperature and atmospheric pressure. Weather stations are ordered according to their elevation, aiming at displaying a pseudo–vertical distribution of the observed quantities.

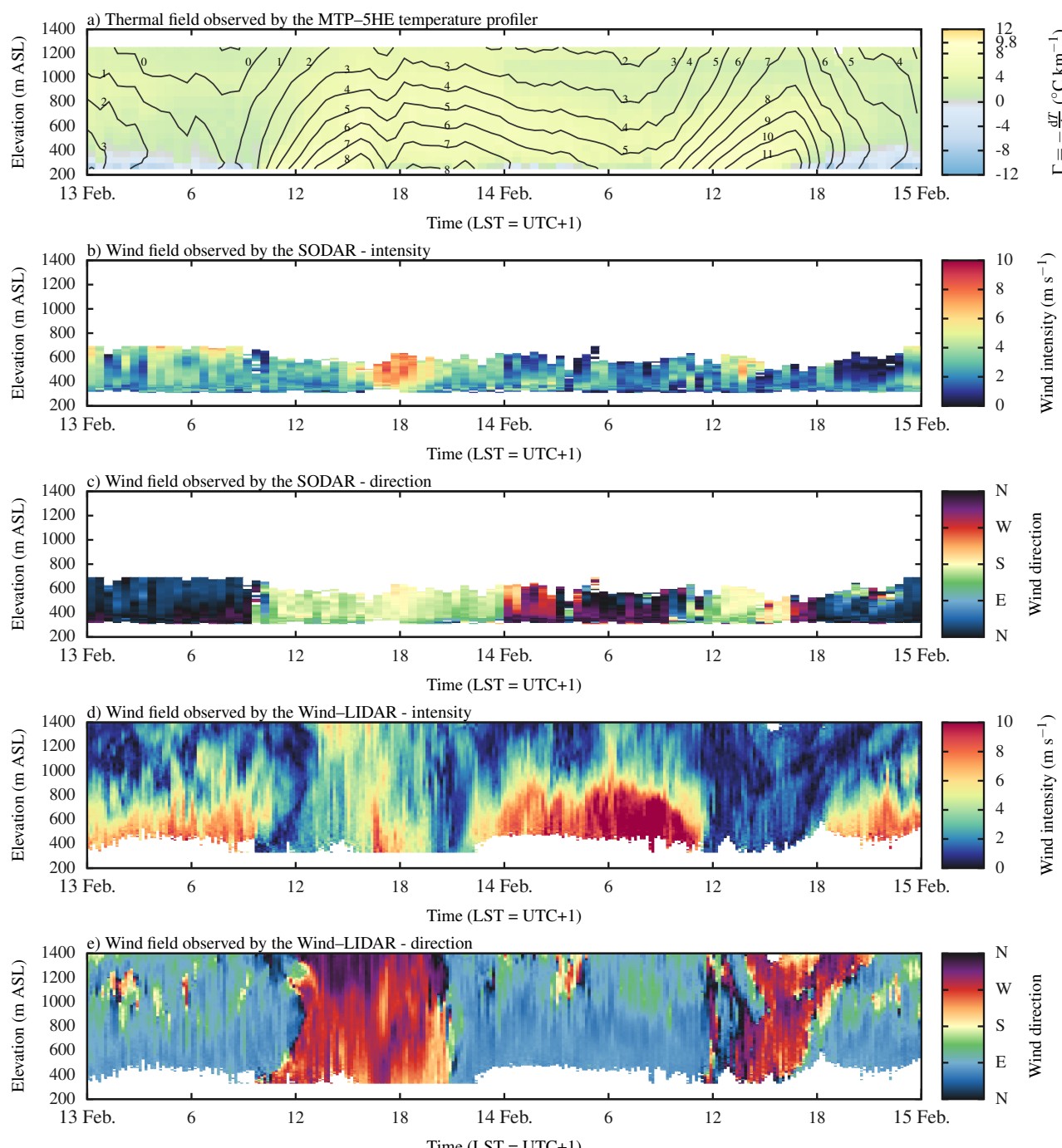

**Figure 7.** Time–height distribution of the thermal field (a) and of the wind fields measured by the SODAR (b–c) and by the Doppler Wind–LIDAR (d–e) inside the Bolzano basin.





**Table 4.** Geographical coordinates of the sampling points (SP) along with number of samples of ambient air collected during the first release of BTEX (VB: vacuum–filled glass bottle; TB: PVF bags). The numbers close to VB and TB indicate the duration in minutes of the sampling.

| 1st release | | | | | | | | | | |
|---|---|---|---|---|---|---|---|---|---|---|
| ID | Latitude | Longitude | Altitude | | | | Samples | | | |
| | [N] | [E] | [m ASL] | VB60 | VB20 | TB25 | TB15 | TB10 | TB5 | Total |
| SP01 | 46.49 | 11.33 | 248 | 1 | - | - | - | - | - | 1 |
| SP02 | 46.48 | 11.32 | 250 | 2 | - | - | - | - | - | 2 |
| SP03 | 46.46 | 11.31 | 238 | 2 | - | - | - | - | - | 2 |
| SP04 | 46.48 | 11.33 | 249 | 2 | - | - | - | - | - | 2 |
| SP05 | 46.45 | 11.33 | 232 | 1 | - | - | - | - | - | 1 |
| SP06 | 46.43 | 11.34 | 320 | 1 | - | - | - | - | - | 1 |
| SP07 | 46.41 | 11.32 | 228 | 1 | - | - | - | - | - | 1 |
| SP08 | 46.46 | 11.32 | 236 | 2 | - | - | - | - | - | 2 |
| SP09 | 46.47 | 11.31 | 358 | 2 | 1 | - | - | - | - | 3 |
| SP10 | 46.46 | 11.30 | 413 | 1 | 1 | - | - | - | - | 2 |
| SP11 | 46.47 | 11.32 | 240 | - | 3 | - | - | - | - | 3 |
| SP12 | 46.50 | 11.30 | 242 | - | 1 | - | - | - | - | 1 |
| SP13 | 46.50 | 11.35 | 275 | - | 1 | 1 | 2 | - | - | 4 |
| SP14 | 46.47 | 11.33 | 263 | - | 3 | - | - | - | - | 3 |
| Total | | | | 15 | 10 | 1 | 2 | - | - | 28 |

## 5.2 Dispersion patterns

The two releases of $SF_6$ were carried out to investigate the fall–out areas of the tracer under different conditions of atmospheric stability and blowing winds, according to the release strategy (Sec. 2.2.2) and the sampling strategy (Sec. 2.2.4) described above. These data can be used to evaluate the spatial patterns of the tracer in the target area and its evolution in 14 different points. Table 3 summarizes the main characteristics of these releases. In particular, Table 3 reports the timetables and the masses of the tracer released, the metadata concerning the incinerator smoke, the number of sampling teams and the number of collected samples. Further, information regarding the coordinates of the sampling points and the collected samples (sampling devices and time resolution) are reported in Tab. 4 and Tab. 5 for the first and the second release, respectively. During both the releases, instantaneous samples of ambient air were also collected outside the laboratory, located in the Bolzano city center, and immediately analyzed as described in Sec. 3.2. These measurements allowed monitoring in quasi–real–time the presence of the tracer in the atmosphere and therefore controlling the effectiveness of the releases.

Figure 8 provides a graphical overview of the dataset, by representing the time evolution of the concentrations measured by the sampling teams during each release. In particular, in view of describing the spatial patterns of the tracer, the sampling

**Table 5.** Geographical coordinates of the sampling points (SP) along with number of samples of ambient air collected during the second release of BTEX (VB: vacuum–filled glass bottle; TB: PVF bags). The numbers close to VB and TB indicate the duration in minutes of the sampling.

| 2$^{st}$ release | | | | | | | | | |
|---|---|---|---|---|---|---|---|---|---|
| ID | Latitude | Longitude | Altitude | | | Samples | | | |
| | [N] | [E] | [m ASL] | VB60 | VB20 | TB25 | TB15 | TB10 | TB5 | Total |
| SP01 | 46.49 | 11.33 | 246 | 2 | - | - | - | - | - | 2 |
| SP02 | 46.48 | 11.32 | 250 | 2 | - | - | - | - | - | 2 |
| SP03 | 46.46 | 11.31 | 238 | 1 | - | - | - | - | - | 1 |
| SP04 | 46.49 | 11.31 | 247 | 2 | - | - | - | - | - | 2 |
| SP05 | 46.45 | 11.33 | 232 | - | 1 | - | - | - | - | 1 |
| SP06 | 46.43 | 11.34 | 320 | - | 1 | - | - | - | - | 1 |
| SP07 | 46.49 | 11.29 | 242 | 1 | - | - | - | - | - | 1 |
| SP08 | 46.48 | 11.31 | 358 | 2 | 1 | - | - | - | - | 3 |
| SP09 | 46.48 | 11.32 | 245 | 2 | - | - | - | - | - | 2 |
| SP10 | 46.50 | 11.36 | 286 | - | 5 | - | - | - | - | 5 |
| SP11 | 46.50 | 11.30 | 242 | 2 | - | - | - | 3 | 5 | 10 |
| SP12 | 46.50 | 11.35 | 275 | - | 3 | - | 2 | 5 | - | 10 |
| SP13 | 46.49 | 11.34 | 254 | 2 | - | - | - | - | 6 | 8 |
| SP14 | 46.47 | 11.33 | 263 | 3 | - | - | - | - | - | 3 |
| Total | | | | 19 | 11 | - | 2 | 8 | 11 | 51 |

points are ordered according to their latitude, *i.e.* from South (below) to North (above), while the horizontal black line marks the latitude of the incinerator.

The first release of BTEX was carried out in the early morning, in order to evaluate the dispersion of pollutants in a stable nocturnal atmosphere with a light down–valley wind (*i.e.* blowing from North to South, see for example the weather station at Bronzolo (WS4) in Fig. 6a and b). In this case, the valley–exit jet from the Isarco Valley, being confined north of the incinerator within the Bolzano basin, did not interact with the released smokes. These atmospheric conditions suggested to envisage a weak dispersion of the tracer and a fall–out area south of the incinerator, in the Adige Valley. The tracer release started at 07:00 LST and ended 1 h later, whereas the sampling activity started at 08:30 LST and continued until 10:45 LST. During this release, the 14 sampling teams collected a total of 28 samples of ambient air in the points represented in the maps in Fig. 9. In particular, Figure 9 shows the position of the sampling teams and the tracer concentrations at four different times (arrows in Fig. 8). At the beginning of the sampling period (45 min after the end of the release, Fig. 9a), only one team north-east of the incinerator observed a concentration of $SF_6$ significantly different from the background (0.03–0.1 ppb), whereas at 09:15 LST the tracer



**Figure 8.** Tracer concentrations measured on 14 February 2017 after each release. Air samples were collected by means of vacuum–filled glass bottles (large rectangles), PVF bags (small rectangles). In addition, instantaneous samples of air (circles) were analyzed. The sampling points are listed from North to South, whereas the two horizontal black lines indicate the latitude of the incinerator. The arrows at the top and bottom of the graph indicate reference time stamps [LST] for Fig. 9 and Fig. 10.

was observed south of the incinerator in the Adige Valley, with concentrations ranging between 0.1 and 0.5 ppb (Fig. 9b). These findings confirm that the transport of the tracer due to the advection by the mean wind dominated over turbulent dispersion, and therefore the fall–out of the tracer at surface level was slower. After 09:30 LST, thanks to the heating of the ground and the

onset of a weak up–valley wind (WS3 and WS4 in Fig. 6a and b), tracer concentrations increased also in the area surrounding the incinerator (Fig. 9c and d), where a maximum concentration of 1.19 ppb was observed (Fig. 9d). However, coherently with the observed wind speed and direction, the tracer never reached the town of Bolzano.

The second release of BTEX, instead, started at 12:45 LST and ended at 14:15 LST, in order to investigate the dispersion processes under a weakly unstable atmosphere and an up–valley wind rising the Adige Valley (see WS3, WS4, WS8, WS11



**Figure 9.** Spatial distribution of the tracer concentrations measured during the release performed in the morning of 14 February 2017.

in Fig. 6a and b). Under these atmospheric conditions, the mixing is usually more efficient, and a faster fall–out of the tracer, north of the incinerator, was expected. Therefore, samples of ambient air were collected starting from 13:10 LST (still during

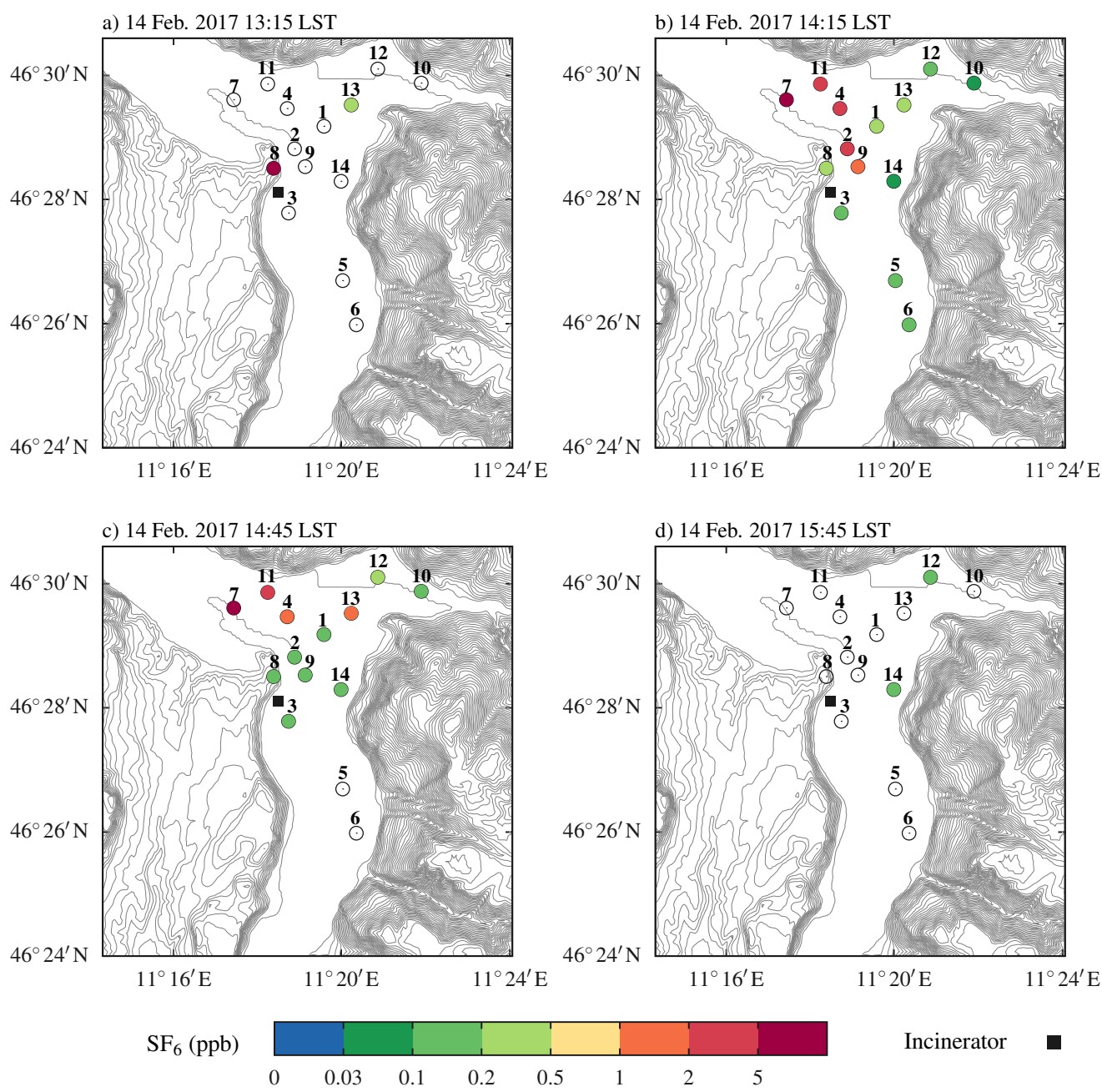

**Figure 10.** Spatial distribution of the tracer concentrations measured during the release performed in the afternoon of 14 February 2017.

the release), until 16:30 LST. Indeed, a concentration of $11.96\,\mathrm{ppb}$ was observed north of the incinerator at 13:15 LST (Fig. 10a), *i.e.* 30 min after the beginning of the release. At the end of the release (14:15 LST, Figure 10b), the up–valley winds

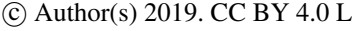


moved the smokes of the plant northward and significant concentrations of $SF_6$ were measured west of Bolzano, inside the

basin, while trace concentrations were observed south of the incinerator. After $30\,min$ (Figure 10c), the tracer spread into the

basin, even if high concentrations were still observed west of Bolzano. Background concentrations of $SF_6$ ranging between 0.2

and $0.5\,ppb$, as shown in Fig. 10a, were also found, as a consequence of the persistence of the tracer released in the morning.

At the end of this second release a total of 51 samples were collected by means of both vacuum–filled glass bottles and PVF

bags.

## 6 Conclusions and outlook

The Bolzano Tracer EXperiment (BTEX) represents one of the few experiments available in the literature performed over com-

plex mountainous terrain to evaluate dispersion processes by means of controlled tracer releases and under well documented

meteorological conditions. The dataset presented in this paper is the result of a 3–year project during which a great effort was

made to properly design, organize and manage all the activities connected to the measurement campaigns. Indeed, preliminary

investigations were performed to identify the most suitable monitoring network (Sec. 3.1) able to properly capture the local

circulation patterns characterizing the meteorology of the Bolzano basin, also by performing *ad hoc* measurement campaigns.

At the same time, the release of the tracer through the stack of the incinerator was designed. This task included: (*i*) the selection

of the tracer (Sec. 2.2.1) and of a representative day for the study area (Sec. 2.2.3); (*ii*) the definition of the injection of the

tracer, to ensure and monitor the steadiness of the release (Sec. 2.2.2); (*iii*) the definition of the devices used to collect samples

of ambient air at ground level (Sec. 2.2.4) and of the required laboratory analysis to measure the tracer concentrations (Sec.

3.2).

On 14 February 2017 two tracer releases were performed. The collected dataset contains 79 samples of tracer ground

concentrations. These concentrations were collected during each release in 14 different locations of the study area and at

different times, thus allowing to evaluate the space–time variability of the dispersion processes. The dataset is completed with

a detailed description of the meteorological field, provided by 15 ground weather stations, one microwave temperature profiler,

one SODAR and one Doppler Wind–LIDAR. In particular, the meteorological data cover a period of $48\,h$ starting from 13

February 2017 00:00 LST, in order to provide a more complete description of the meteorological processes within the study

area.

The uniqueness of BTEX makes the collected dataset a useful benchmark for testing dispersion models in complex terrain.

## 7 Data availability

All data presented in this paper are publicly available at the World Data Center PANGAEA: https://doi.pangaea.de/10.1594/

PANGAEA.898761 (Falocchi et al., 2019).



*Author contributions.* MF led the writing of the manuscript, and prepared all figures and tables. WT wrote the section on chemical measurements. DZ revised preliminary versions of the manuscript. All authors contributed to suggest ideas and reviewing the manuscript.

All Authors contributed to the design and planning of the experiment and participated in the field campaigns. WT provided the required tools for air samples collection, instructed the team of observers and performed all the laboratory analyses. MF, LG, ET and GA managed the SODAR and the Doppler Wind–LIDAR operations. GA, ET and LG contributed to set-up the modeling chain and operated it during the field campaigns. MF collected all the data, both from the field experiment and from permanent observational facilities in the target area and surroundings, organized the dataset, and created its publication on World Data Center PANGAEA. DZ, as Principal Investigator, managed the

overall execution of the project, including partners' involvement and commitment, from the initial concept to the conclusion, and suggested the preparation of the present paper.

*Competing interests.* The authors declare that they have no conflict of interest.

*Acknowledgements.* The authors acknowledge Marco Palmitano (Eco–Center s.p.a.) and Bruno Eisenstecken (Eco–Center s.p.a.) who encouraged and strongly supported this study, also by renting the Doppler Wind–LIDAR. The authors are also grateful to all the personnel

of Eco–Center s.p.a., Eco–Research s.r.l., of the Environmental Agency of the Autonomous Province of Bolzano and of the "Mario Negri Institute" for participating to the measurements activities during the releases. The Meteorological Office of the Autonomous Province of Bolzano is acknowledged for kindly providing data from their weather stations. The Environmental Agency of the Autonomous Province of Bolzano, Massimo Guariento and Luca Verdi are kindly acknowledged for data from the microwave temperature profiler.



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
