# Peer review of "A dataset of tracer concentrations and meteorological observations from the Bolzano Tracer EXperiment (BTEX) to characterize pollutant dispersion processes in an Alpine valley."

_Earth System Science Data, 2019_

## Referee Comment (RC1) · Anonymous Referee #1 · 9 Sep 2019

The paper describes an experimental campaign carried out in an alpine valley. The measurements include both meteorological and concentration data. The measurement techniques and strategy are shown in details as well as the gathered data-sets. Observations of meteorological quantities and tracer concentrations are very useful for both understanding the flow and turbulence dynamics and evaluating the performances of the numerical models, especially in complex terrain. They are particularly interesting and useful when available for all the scientific community. For these reasons a strongly support the publication of the paper. I have only minor comments and suggestions for

the authors.

Line 39: TRACT experiments might be cited here.

Lines 111-115: It is not clear whether two different kinds of system were used and two different systems to fill them were adopted.

Line 123: I wonder if "predicted" would be better than "foreseen"

Line 130: NWP simulations at a resolution of 200 m are not standard. This resolution, although justified by the need to describe small scales in the complex terrain, falls in the "grey zone" of the atmosphere. Thus, this point deserves some discussion.

Table 2: It may be useful to write which kind of instrument the information given in the table refer to (not only the technical name)

Section 3.2: I wonder whether, for reasons of clarity, the description of the instrumentation for measuring the SF6 concentrations, bottles and bags, might be moved here.

Line 202: Add MTP to indicate the instrument utilised for the measurements.

Line 213: Please, indicate which kind of data are used to plot the geopotential height map.

Line 269: Substitute Figure with Fig. as for the others.

Figure 8: I have some concerns about this figure. It does not show the positions of the sampling points along the west-est direction. Further, since some of the samplings last less than others, it seems that the concentration would be zero at some time which, instead, might be not true.

Line 277: The text refers to Bolzano which is not shown on the maps.

Line 285: It seems that concentrations were found upwind to the incinerator.

Conclusions: Nothing is said about the strategy adopted to locate the samplings. Did it succeed? Did the model correctly predict the plume dispersion helping to properly

positioning bottles and bags? Were numerical simulations repeated and the results compared with the observations?

Line 309: Were simulations done and compared with the measured data?

———————————————

---

## Referee Comment (RC2) · Anonymous Referee #1 · 4 Oct 2019

The answers of the authors to my comments are satisfactory. The paper can be published in the present form

---

## Author Comment (AC1) · 4 Oct 2019

**Reply to the remarks of Referee #1 (R1).**

In the following lines, text in red was removed, while corrections are written in blue.

**R1**: **Line 39: TRACT experiments might be cited here.**

**Reply**: We thank the Referee for the suggested reference.

[Figure]

**Action**: References to the suggested project are added into the paper.

**R1**: **Lines 111–115: It is not clear whether two different kinds of system were used and two different systems to fill them were adopted.**

**Reply**: During BTEX samples of ambient air were collected by adopting two different technologies: (1) vacuum–filled glass bottles and (2) polyvinyl fluoride (PVF) bags. Bottles, with a capacity of 1 liter were filled by means of valves equipped with calibrated nozzles. Each nozzle allowed to fill a bottle at a constant inflow within a fixed time–frame. Nozzles with characteristic filling–times of either 20 or 60 min were used. The filling of the PVF bags, instead, was regulated by a suitable air pump controlling the inflow speed.

**Action**: The text is modified as follows:
"After each release, samples of ambient air were collected by means of two different technologies: vacuum–filled glass bottles and polyvinyl fluoride (PVF) bags (hereafter PVF bags)."

**R1**: **Line 123: I wonder if "predicted" would be better than "foreseen".**

**Action**: The text is modified as follows:
"For these reasons, seven sampling teams were located in the main residential areas, where most of the receptors are concentrated, while the remaining ones were placed in the surroundings of the incinerator, in agreement with the ground concentrations  predicted by an operational modeling chain composed of both meteorological and air quality models, specifically set–up for the present field campaign."

**R1**: **Line 130: NWP simulations at a resolution of 200 m are not standard. This resolution, although justified by the need to describe small scales in the complex terrain, falls in the "gray zone" of the atmosphere. Thus, this point deserves some discussion.**

**Reply**: NWP simulations were run with the prognostic model WRF (Weather Research and Forecasting model) with a horizontal resolution of 500 m in the innermost domain. Then the CALMET model was used as a "physically–based interpolator" to further increase the horizontal resolution from 500 m to 200 m. CALMET is a diagnostic mass–consistent model and does not solve the full dynamic equations controlling the atmospheric flows. We agree that the horizontal resolution adopted for the NWP simulations (*i.e.* 500 m) can fall, in principle, in the so–called gray zone or "terra incognita", *i.e.* those scales where turbulence is neither sub–grid nor fully resolved, but rather is partially resolved. Then the use of a PBL parameterization can be questionable. However, this rather high resolution was necessary to capture adequately the spatial variability of meteorological fields in the Bolzano basin. On the other hand, given also the fact that for most of the simulated period the atmospheric boundary layer was stably stratified, this resolution does not allow to fulfill the requirements for running Large Eddy Simulations, *i.e.* explicitly resolve the largest and most energetic scales of turbulence. In this case a horizontal resolution of few tens of meters should be probably adopted to resolve the most energetic eddies (see Cuxart 2015 for a discussion on this topic). For computational reasons the adoption of such a resolution was not feasible here.

**Action**: The following lines are added at the end of Sec. 2.2.4:
"The resolution adopted for the numerical weather prediction simulations (*i.e.* 500 m) can fall, in principle, in the so–called gray zone or "terra incognita", *i.e.*

those scales where turbulence is neither sub–grid nor fully resolved, but rather is partially resolved. However, this rather high resolution was necessary to capture adequately the spatial variability of meteorological fields in the Bolzano basin. The CALMET model, instead, was used as a "physically–based interpolator" to further increase the horizontal resolution from 500 m to 200 m."

**R1**: **Table 2: It may be useful to write which kind of instrument the information given in the table refer to (not only the technical name).**

**Action**: The caption of the Tab. 2 was modified as follows:
"Technical characteristics of the MTP–5HE Microwave Temperature Profiler (MTP–5HE) installed at the airfield of Bolzano."

**R1**: **Section 3.2: I wonder whether, for reasons of clarity, the description of the instrumentation for measuring the SF$_6$ concentrations, bottles and bags, might be moved here.**

**Action**: As suggested, the first paragraph of Sec. 2.2.4 is moved to Sec. 3.2.

**R1**: **Line 202: Add MTP to indicate the instrument utilized for the measurements.**

**Action**: The sentence was modified as follows:
"Temperature profiles collected every 10 min and measured by means of the Microwave Temperature Profiler (MTP–5HE) every 10 min. These data are provided by the Physical Chemistry Laboratory of the Environmental Agency of the Autonomous

Province of Bolzano."

**R1**: **Line 213: Please, indicate which kind of data are used to plot the geopotential height map.**

**Reply**: The maps of geopotential height, reported in Fig. 5, were freely downloaded from the web–site: `http://www.wetterzentrale.de`. As reported in the caption of Fig. 5, these maps represent an output of the reanalyzes from the Climate Forecast System (CFS) model.

**Action**: The text was modified as follows:
"Releases of $SF_6$ were performed on 14 February 2017, when over North Italy a high–pressure system was present with very weak synoptic winds (see maps of geopotential height at 500 hPa in Fig. 5). The evolution of the synoptic conditions from 13th February 00:00 UTC to 15th February 00:00 UTC is reported in Fig. 5, in terms of maps of geopotential height at 500 hPa as simulated by the reanalyzes of the Climate Forecast System (CFS) model (source: `www.wetterzentrale.de`)."

**R1**: **Line 269: Substitute Figure with Fig. as for the others.**

**Action**: Correction implemented throughout the paper.

**R1**: **Figure 8: I have some concerns about this figure. It does not show the positions of the sampling points along the west–est direction. Further, since some of the samplings last less than others, it seems that the concentration would be zero at some time which, instead, might be not**

**true.**

**Reply**: Figure 8 provides a graphical representation of the dataset of the available concentrations. The figure displays, for each release (gray area), the position of the sampling teams in the North–South direction, along the axis of the Adige Valley. The displacement of the sampling points in the cross–valley direction, instead, was not represented in the Figure because less relevant, being the wind aligned with the valley axis. The colored boxes, instead, are used to identify the samples of ambient air and the measured concentrations. The length of the box fits the duration of the sampling, its height allows to distinguish the sampling method (bottles or bags), whereas the color indicates the concentration of the tracer as measured after the laboratory analyzes. Blue boxes correspond to samples in which the tracer concentration was lower than the sensitivity of the analyses, *i.e.* $30\,\mathrm{ppt_v}$. White areas, instead, correspond to periods without samplings. Therefore, time periods with no appreciable concentration (with measurements, but below the detectability limit) are represented by the blue boxes, while periods without measurements are represented by the white areas.

**Action**: The following sentence is introduced in the text after line 260:
"Figure 8 provides a graphical overview of the dataset, by representing the time evolution of the concentrations measured by the sampling teams during each release (gray areas). In particular, in view of describing the spatial patterns of the tracer along the axis of the Adige Valley, *i.e.* in agreement with the observed wind regime, the sampling points are ordered according to their latitude, *i.e.* from South (below) to North (above), while the horizontal black line marks the latitude of the incinerator. Each box corresponds to one sample: the length of the box fits the timing and the duration of the sampling, whereas the color corresponds to the measured concentration. In particular, blue boxes correspond to samples in which the tracer concentration was lower than the detectability limit of the laboratory analyzes, *i.e.* $30\,\mathrm{ppt_v}$. White areas, instead, correspond to periods

without samplings."

**R1**: **Line 277: The text refers to Bolzano which is not shown on the maps.**

**Reply**: The city of Bolzano is indicated in Fig. 1 as BZ.

**Action**: In order to make maps more readable, in Fig. 9 and 10, we added the same acronym (BZ) also in these Figures.

**R1**: **Line 285: It seems that concentrations were found upwind to the incinerator.**

**Reply**: During the second release, tracer concentrations south of the incinerator were attributed to the stagnation and the recirculation of the tracer close to the valley floor, after the first release (07:00–08:00 LST). In Fig. 10a, no concentrations are observed upwind of the incinerator because the sampling was not active in that area at 13:15 (as also displayed in Fig. 8). Instead, concentrations of the tracer were observed at the sampling points SP03, SP05 and SP06 after 14:00.

The reconstruction of the meteorological processes during the experimental activities and the consequent modeling of the tracer dispersion performed by means of numerical simulations, detailed in Tomasi *et al.* (2019), confirmed this hypothesis and the simulated patterns of tracer concentrations at the ground level were found to be in agreement with observations. Figure 6 in Tomasi *et al.* (2019) compares the ground concentrations of the tracer at different times (rows), as provided by the dispersion models used in that study, *i.e.* CALPUFF (left column) and SPRAYWEB (center and right column) with two different parameterizations. In particular, in the last row (at 14:00 LST, during the second

release) all models simulated a residual concentration of tracer south to the incinerator.

**R1**: **Conclusions: Nothing is said about the strategy adopted to locate the sam-plings. Did it succeeds? Did the model correctly predict the plume dis-persion helping to properly positioning bottles and bags? Were numerical simulations repeated and the results compared with the observations?**

**R1**: **Line 309: Were simulations done and compared with the measured data?**

**Reply**: As reported in the Section entitled "Sampling strategy", sampling points were selected on the basis of two different needs:

1. providing a direct measure of the tracer concentrations in the main residen-tial areas, where most of the receptors are concentrated, regardless of the dynamics of the plume from the incinerator;

2. testing the reliability of a modeling chain composed of both meteorological and air quality models. In order to satisfy this need, the sampling points were located in the surroundings of the incinerator, in agreement with the ground concentrations predicted by the modeling chain. Moreover, our fa-miliarity with the typical meteorological processes characterizing the study area gained during previous studies was much helpful in the interpretation of the simulated scenarios and, therefore, in the positioning of the sampling points.

In most of the collected samples of ambient air tracer concentrations were found, thus indicating that the adopted sampling strategy was quite appropriate and also that the modeling chain reproduced quite well the fall–out area of the tracer. In particular, the patterns of ground concentrations simulated by the modeling chain

were found to be in agreement with the observed meteorological fields, especially during the first release, in the early morning. The fall–out area of the tracer simulated during the second release, in the afternoon, instead, was a little bit more uncertain. Indeed, the modeling chain anticipated the onset of the down–valley wind of the Adige Valley. In this case, the sampling points were located on the basis of the observed wind field and of the expertise gained by the modeling team from previous tests.

After the experimental activities, the dispersion of the released tracers was numerically simulated at high resolution. In particular, the meteorological field was simulated by means of the WRF model, whereas the dispersion of the tracer was simulated by means of the CALPUFF model and by the SPRAYWEB model. A detailed description of the models setup and of the simulations is provided in Tomasi *et al.* (2019).

**Action**: The following lines are added to the conclusions:
On 14 February 2017 two tracer releases were performed. The collected dataset contains 79 samples of tracer ground concentrations,. These concentrations were collected during each release in 14 different locations of the study area and at different times, thus allowing to evaluate the space–time variability of the dispersion processes. The complex orography of the study area and its related heterogeneous meteorological fields did not allow a regular distribution of the sampling points around the incinerator, *e.g.* in concentric circles. Therefore, seven sampling points were dislocated in the main residential areas, in the neighboring of the incinerator, while the other seven sampling points were placed in agreement with the fall–out areas of the tracer, as predicted by a modeling chain, specifically setup for the purposes of the study and composed of both meteorological and dispersion models. In particular, for a proper interpretation of the output of the modeling chain, a key role was played by the expertise gained by the modeling team on the typical meteorological processes characterizing the study area. The adopted strategy allowed

to capture the space–time variability of the tracer at ground level. More details on the numerical experiments carried out to simulate the tracer dispersion can be found in Tomasi *et al.* (2019).

The dataset is completed with a detailed description of the meteorological field, provided by 15 ground weather stations, one microwave temperature profiler, one SODAR and one Doppler Wind–LIDAR. In particular, the meteorological data cover a period of 48 h starting from 13 February 2017 00:00 LST, in order to provide a more complete description of the meteorological processes within the study area.

Additional references:

- Cuxart J. When can a high–resolution simulation over complex terrain be called LES?. *Front. Earth Sci.* 2015, 3, 87.

- Tomasi, E., Giovannini, L., Falocchi, M., Antonacci, G., Jiménez, P.A., Kosovic, B., Alessandrini, S., Zardi, D., Delle Monache, L. and Ferrero, E. Turbulence parameterizations for dispersion in sub–kilometer horizontally non–homogeneous flows. *Atmospheric Research*, 2019, 228: 122-136.

---

## Referee Comment (RC3) · Anonymous Referee #2 · 9 Jan 2020

This paper excellently describes a tracer experiment in the mountains of Italy, to study the fallout from an incinerator in a valley with high terrain. The experiment was clearly very carefully designed and a great deal of work went into ensuring that measurement sites were selected intelligently (using model runs to predict wind patterns), and that the execution was coordinated very well, with a team of 14 scattered throughout the area to take simultaneous measurements.

The results are presented very clearly, with helpful maps to assist in visualizing the dispersion of the tracer over the hours following each release.

The paper is well-written and very easy to understand. Recommend publication in its current form.